# Botulinum Toxin for Axial Postural Abnormalities in Parkinson’s Disease: A Systematic Review

**DOI:** 10.3390/toxins16050228

**Published:** 2024-05-15

**Authors:** Marialuisa Gandolfi, Carlo Alberto Artusi, Gabriele Imbalzano, Serena Camozzi, Mauro Crestani, Leonardo Lopiano, Michele Tinazzi, Christian Geroin

**Affiliations:** 1Department of Neurosciences, Biomedicine and Movement Sciences, University of Verona, 37134 Verona, Italy; serena.camozzi@studenti.univr.it (S.C.); mauro.crestani@univr.it (M.C.); 2Neuromotor and Cognitive Rehabilitation Research Centre (CRRNC), University of Verona, 37134 Verona, Italy; 3Neurorehabilitation Unit, AOUI Verona, 37134 Verona, Italy; 4Department of Neuroscience “Rita Levi Montalcini”, University of Turin, 10126 Turin, Italy; caartusi@gmail.com (C.A.A.); gabriele.imbalzano@unito.it (G.I.); leonardo.lopiano@unito.it (L.L.); 5SC Neurology 2U, AOU Città della Salute e della Scienza, 10126 Turin, Italy; 6Department of Surgery, Dentistry, Paediatrics and Gynecology, University of Verona, 37134 Verona, Italy; christian.geroin@univr.it

**Keywords:** axial postural abnormality, Parkinson’s disease, botulinum toxin, physiotherapy, disability, pain

## Abstract

Axial postural abnormalities (APAs), characterized by their frequency, disabling nature, and resistance to pharmacological treatments, significantly impact Parkinson’s disease and atypical Parkinsonism patients. Despite advancements in diagnosing, assessing, and understanding their pathophysiology, managing these complications remains a significant challenge. Often underestimated by healthcare professionals, these disturbances can exacerbate disability. This systematic review assesses botulinum toxin treatments’ effectiveness, alone and with rehabilitation, in addressing APAs in Parkinson’s disease, utilizing MEDLINE (PubMed), Web of Science, and SCOPUS databases for source material. Of the 1087 records retrieved, 16 met the selection criteria. Most research has focused on botulinum toxin (BoNT) as the primary treatment for camptocormia and Pisa syndrome, utilizing mostly observational methods. Despite dose and injection site variations, a common strategy was using electromyography-guided injections, occasionally enhanced with ultrasound. Patients with Pisa syndrome notably saw consistent improvements in APAs and pain. However, studies on the combined effects of botulinum toxin and rehabilitation are limited, and antecollis is significantly under-researched. These findings recommend precise BoNT injections into hyperactive muscles in well-selected patients by skilled clinicians, avoiding compensatory muscles, and underscore the necessity of early rehabilitation. Rehabilitation is crucial in a multidisciplinary approach to managing APAs, highlighting the importance of a multidisciplinary team of experts.

## 1. Introduction

Axial postural abnormalities (APAs) affect over 20% of individuals diagnosed with Parkinson’s disease (PD), which significantly impairs their quality of life [1,2,3,4,5,6,7,8,9]. In addition, these abnormalities are closely associated with increased pain, a higher risk of falls, and reduced life quality [3,10,11,12,13,14,15,16,17,18]. Axial symptoms are predominantly observed in the middle to advanced stages of PD and are characteristically resistant to conventional dopaminergic therapies [19,20,21,22,23]. Recent advancements have led to a better understanding of their prevalence and classification. Traditionally, APAs were classified into four categories based on misalignment type and direction (camptocormia, Pisa syndrome, antecollis, and scoliosis) [2,24,25]. However, this classification has been refined with updated harmonization of terminology and criteria, as recommended by the Movement Disorder Society (MDS) Task Force on Postural Abnormalities. Specifically, camptocormia (CC), previously regarded as a singular condition characterized by sagittal plane trunk flexion, has been reclassified into two distinct types based on the fulcrum location; camptocormia with a “lumbar fulcrum” involves a forward trunk flexion of >30° originating from the spinous processes of L1 to the sacrum and hip, whereas camptocormia with a “thoracic fulcrum” entails an anterior trunk flexion exceeding 45°, spanning from C7 to T12-L1 [26]. Antecollis is defined by an anterior neck flexion exceeding 45°, and Pisa syndrome (PS) is identified by a lateral trunk inclination exceeding 10° [26]. To describe milder APAs, recommendations include lateral trunk flexion (≥5° to ≤10°), anterior trunk flexion with upper (≥25° to ≤45°) and lower (>15° to ≤30°) fulcrum, and “anterior neck flexion” (>35° to ≤45°) are recommended [26].

Despite these advancements in consensus-based classification and assessment, the pathophysiological mechanisms driving APAs in PD still need to be completely understood, limiting our knowledge on the development of effective treatment strategies [19,26,27,28,29,30,31,32,33,34]. The comprehension of APAs pathophysiology in PD is partial, with two main hypothesized mechanisms involving central (e.g., imbalance in basal ganglia functioning leading to dystonia/rigidity, higher cognitive function deficits, and altered sensory-motor integration) and peripheral factors such as myopathy, and alterations in the spine or soft tissues [2,35,36,37,38,39,40,41,42,43,44,45,46,47,48,49,50,51,52,53,54,55,56,57,58,59,60,61,62,63,64,65]. The literature emphasizes the importance of formulating APAs management strategies based on the disturbance etiopathogenesis to target contributing factors in therapy [19,34,66,67].

Most electromyography (EMG) studies investigating the pathophysiology of these disturbances have been conducted on a cross-sectional level [24,50]. They have yet to yield consistent findings regarding muscle activation patterns in the axial muscles across different APAs. However, there appears to be a discernible trend suggesting that central mechanisms, possibly represented by dystonic patterns, may play a crucial role in developing these conditions [2,68,69,70].

The role of EMG study in these conditions can now be considered for evaluating hyperactive muscles, which can be possible targets of botulinum toxin (BoNT) when not considered compensatory [19,71]. Given the potential contribution of dystonic patterns to APAs in PD [2,68,69,71], and more in general, the possible recognition of hyperactivity of axial muscles, BoNT emerges as a promising therapeutic intervention in PD for the management of APAs [72,73,74,75,76,77]. Beyond its primary mechanism of action, which involves inhibiting acetylcholine release at the neuromuscular junction, thereby reducing muscle hyperactivity, BoNT also significantly affects non-motor symptoms, most notably pain [78,79]. Pain is a prevalent and debilitating non-motor symptom in PD, often exacerbated by muscular rigidity, dystonia, and abnormal postures [13,14,80,81].

In this scenario, a systematic review was conducted to examine the potential pharmacological BoNT injection treatments for APAs in PD based on the new expert classification of APAs in Parkinsonism. We consider this review a critical step toward enhancing the clinical management of APAs in PD using BoNT. By aligning treatment approaches with the latest expert-based classification and emphasizing the need for further research, our work aimed to improve patient care and better understanding of the mechanisms underlying APAs in PD.

## 2. Results

Following the selection process, we included 16 records involving 104 patients in the review, as shown in Figure 1. The main data from these included studies are presented and summarized in Table 1.

The majority of the studies (87.5%, n = 14) involved stand-alone BoNT for 77 patients (74%). Only two studies (12.5%) combined BoNT treatment with rehabilitation, covering 27 patients (26%).

In terms of quality, most studies were assessed as being of poor quality (n = 6) [35,82,83,84,85,86]. However, two studies were rated as good quality [87,88] and five as fair [89,90,91,92,93]. Detailed information from the included studies is reported in Appendix A.

**Table 1 toxins-16-00228-t001:** Summary of reports of positive or negative effects of BoNT A (Group A) and BoNT A combined with physical rehabilitation (Group B) for treating APAs in PD.

Type of Axial Postural Abnormality	Principal Outcome Report	Type of Study	Patients Treated	Type of Toxin	Muscles Injected and Dose (U)	QA-LE
**Group A. Studies of BoNT for treating APAs in PD.**						
Antecollis	Positive	2 Case series [82,89]	3	2 OnaBoNT 1 AboBoNT	- Bilat Med Scalene: 50 U - Bilat Lev Scapulae: 25 U - Bilat Longus Colli 40–50 U - Bilat Ant Scalene: 10 U	Poor—4 [82] Fair—4 [89]
	Negative	1 Case series [83]	7	7 AboBoNT	- Uni/Bilat Lev Scapulae: - 15 U or 50–250 U - Uni/Bilat SCM: 50–100 U - Uni/Bilat Splenius capitis: 200 U	Poor—4
Camptocormia	Positive	2 Case series [35,84] 1 Case report [94]	13	13 OnaBoNT	- Bilat Ext Abdominal oblique: 75–90 U, 100–120 U as maintenance - Bilat Rectus Abdominis: 200 U or range 150–400 U - Monolat Ext Abdominal Oblique: 200 U - Paraspinal (200 U or unknown dosage)	Poor—4 [84] NA—4 [94] Poor—4 [35]
	Negative	2 Case series [86,91] 1 Goal attainment-controlled study [85]	15	10 IncoBoNT3 AboBoNT2 OnaBoNT	- Bilat Iliopsoas: 300 U - Bilat Rectus Abdominis: 200 U or 200 ± 63 U - Bilat Iliopsoas: 210 ± 50 U - Unilat or Bilat Iliopsoas: 500 U	Poor—4 [86] Fair—4 [91] Poor—4 [85]
Pisa syndrome	Positive	1 Longitudinal and case-control study [92] 1 Prospective, pilot study [88] 1 Case report [95] 1 Blinded cross-over trial [93]	31	26 OnaBoNT 4 AboBoNT 1 IncoBoNT	- Ipsilat or EMG-hyperactive Longissimus Thoracis: 15–50 U or 50–75 U - Ipsilat or EMG-hyperactive Iliocostalis Lumborum: 15–50 U or 50–75 U - Ipsilat Quadratus Lumborum: 50 U - Ipsilat or EMG-hyperactive Spinalis Dorsi: 50–75 U - Ipsilat or EMG-hyperactive Abdominal Oblique: 50–75 U - Ipsilat or EMG-hyperactive Iliopsoas: 50–75 U	Fair—4 [92] Good—2c [88] NA—4 [95] Fair—4 [93]
	Negative	NA	NA	NA	NA	
Combined Camptocormia and Pisa syndrome	Positive	1 Case report [90]	1	1 OnaBoNT	- Ipsilat Paraspinal: 20–30 U	Fair—4
	Negative	NA	NA	NA	NA	
**Group B. Studies of Combined BoNT and Rehabilitation for treating APAs in PD.**						
Pisa syndrome	Positive	1 RCT [87]	13	13 IncoBoNT	- Bilat Iliopsoas: max 50 U - Ipsilat or contralat Multifidus, Rectus Abdominis, Inferior Thoracic and Lumbar Paravertebral muscles (unspecified): max 50 U per site	Good—1b
	Negative	NA	NA	NA	NA	
Combined Camptocormia and Pisa syndrome	Positive	1 Case report [96]	1	1 AboBoNT	Bilat Paraspinal muscles (unspecified): 600 U across 6 sites	NA—4
	Negative	NA	NA	NA	NA	

Legend: Abbreviations: Ant, Anterior; Bilat, Bilateral; APAs, Axial Postural Abnormalities; BoNT, Botulinum toxin; Contralat, Contralateral; EMG, Electromyography; Ext, External; Ipsilat, Ipsilateral; Lev, Levator; Med, Medial; Monolat, Monolateral; NA, not available; PD, Parkinson’s disease; U, units; RCT, Randomized Controlled Trial; SCM, Sternocleidomastoid; Unilat, Unilateral.

### 2.1. Stand alone Botulinum Toxin Treatment

The studies predominantly addressed CC (or mild anterior trunk flexion) appearing in 43% of the studies (n = 6) and affecting 28 patients [35,84,85,86,91,94].

This was followed by PS (or mild forms of lateral trunk flexion) noted in 28.6% of the studies (n = 4) for 36 patients [88,92,93,95], and AC (or mild forms of anterior neck flexion) in 21.43% of the studies (n = 3) for 12 patients [82,83,89]. There was also a single case study involving both CC and PS [90].

The study designs varied significantly across the reviewed research, reflecting the diverse approaches to investigating APAs. Most studies, comprising 72% (n = 10), were either case series or reports [35,82,83,84,86,89,90,91,94,95], followed by other study designs (7% (n = 1) were goal attainment-controlled studies [85], 7% (n = 1) was a longitudinal observational study and case-control [92], 7% (n = 1) was a blinded crossover trial [93], and 7% (n = 1) was a prospective pilot study [88]) (Figure 2).

### 2.2. Effects in Patients with Camptocormia

PD patients with CC underwent BoNT serotypes A injections, such as abobotulinumtoxin [91], onabotulinumtoxin [35,84,86,90,94], and incobotulinumtoxin [85]. The dosage of BoNT varied among studies and was based on the type of BoNT as well as the injection sites. Three studies used ultrasound-guided BoNT injection [84,85,91], one used CT-guided BoNT injection [86], one EMG-guided technique [94], and two used a blind/unspecified injection technique [35,90]. Bilateral rectus abdominis (n = 4) [35,85,86,94], iliopsoas (n = 3) [85,86,91], external oblique muscle (n = 2) [84,94], and paraspinal muscles (n = 1) [35] were injected.

Moreover, when CC was associated with PS, paraspinal muscles were injected [90]. The efficacy of BoNT was evaluated using several objective and subjective outcome measures. The results varied among studies. Three studies demonstrated a moderate to marked posture improvement following injection in patients with CC [35,84] and in patients with CC combined with PS [90]. In contrast, no improvement in posture after injection was reported in three studies [85,86,91]. Three studies reported significant abdominal pain relief [84,85,94], but in only two studies was it associated with CC severity reduction [84,94]. No other significant improvement in posture, related disability, and quality of life was reported (Appendix A).

### 2.3. Effects in Patients with Pisa Syndrome

PD patients with PS underwent BoNT serotypes A injections, including abobotulinumtoxin [93], onabotulinumtoxin [88,90,92], and incobotulinumtoxin [95]. The dosage of BoNT varied among studies and was based on the type of BoNT as well as the injection sites. One study used ultrasound-guided BoNT injection [88], one used EMG-guided injection [93], and one used EMG plus ultrasonography [92]. One study did not use any specific injection technique [90].

Thoracic and lumbar paraspinal muscles ipsilateral (n = 5) [88,90,92,93,95] or contralateral (n = 1) [88] to the trunk bending side, iliopsoas muscles (n = 1 studies) [88], external/internal oblique muscle (n = 2) [88,92], and quadratus lumborum (n = 1) [95] were injected. Three studies specified the exact paraspinal muscles injected, distinguishing between longissimus dorsi, iliocostalis lumborum, and spinalis dorsi [88,92,95].

The efficacy of BoNT was evaluated using several objective and subjective outcome measures. A moderate to marked PS and pain improvement after injection was highlighted in four studies [88,90,93,95]. No other significant improvement in disability and quality of life was reported (Appendix A).

### 2.4. Effects in Patients with Antecollis

PD patients with AC underwent BoNT serotypes A injections, such as abobotulinumtoxin [82,83] and onabotulinumtoxin [89]. The dosage of BoNT varied among studies and was based on the type of BoNT and the injection sites. EMG-guided BoNT injection was reported in two studies [83,89], while no injection method was reported in one study [82]. Unilateral/bilateral levator scapulae (n = 3) [82,83,89], sternocleidomastoid muscle (n = 1) [83], splenius capitis (n = 1) [83], longus collis (n = 1) [89], and anterior/medial scalene (n = 2) [82,89] were injected. The efficacy of BoNT was evaluated using several objective and subjective outcome measures. Two studies observed a minor improvement in antecollis and pain relief [83,89]. No other significant improvement in disability and quality of life was reported (Appendix A).

### 2.5. Botulinum Toxin Treatment Combined with Rehabilitative Interventions

BoNT treatment combined with rehabilitative interventions was investigated in only two studies, one RCT and one single case study, involving a total of 27 patients [87,96]. The RCT investigated 26 patients with PS randomized to receive either BoNT (experimental group) or saline injections (control group) before participating in the same intensive rehabilitation program [87]. The methodological aspects are detailed in Appendix A. The instrumental analysis of changes in the static lateral trunk flexion via kinematic analysis showed a significant reduction in the severity of PS in the experimental group at the end of the rehabilitation and the 3-month follow-up assessment compared to the control group with a reduction in PS severity at 3-month follow-up of 50%. Group A experienced a greater reduction in pain scores than Group B.

In the single case by Santamato et al. [96], a patient with PS undergoing one single BoNT treatment injection combined with a rehabilitation program (agility exercises, stretching exercises, and Pilates exercises) for five weeks exhibited a regression of most signs and symptoms of PS 15 days after the BoNT treatment. In particular, a PS severity regression of around 20° of the severity of PS and a reduction of four points at the VAS for pain. No studies specifically examined forward neck flexion and antecollis. For details, see Appendix A.

## 3. Discussion

This systematic review of research studies on the effects of BoNT on treating APAs in PD presents a comprehensive overview, reflecting the multifaceted approach to managing these debilitating conditions. Our findings, drawn from various study designs and patient presentations, shed light on the nuanced responses to BoNT treatment across different forms of APAs in PD, including CC, PS, and occasionally AC. Positive and negative outcomes were documented for AC, although the evidence is limited mainly to single-case studies, resulting in a poor quality of evidence [82,83,89]. Similarly, CC showed mixed results; positive effects were observed when abdominal and paraspinal muscles were treated, while negative outcomes emerged from treatments targeting the iliopsoas muscles. Despite these mixed results on motor symptoms, the significant pain reduction highlights the potential of BoNT to alleviate not only motor but also non-motor symptoms (pain) [35,83,84,87,88,89,90,93,94,95,96].

In contrast, studies on PS consistently reported positive outcomes, with the quality of these studies ranging from fair to good [87,88,92,93,95,96]. This suggests that ipsilateral injection treatments may effectively reduce the severity of Pisa syndrome with a more consistent response among studies on PS when compared to those on CC. It might depend on the more focused targeting of muscles involved in lateral trunk flexion and the asymmetric presentation of the disturbance [87,93].

Additionally, paravertebral muscles (frequently treated in PS and less suitable for treating CC, being primarily extensor muscles) are, perhaps, more accessible than muscles targeted for the anterior trunk flexion, especially when considering that many studies did not use ultrasound guidance. However, for PS, the variation in injection sites and techniques, albeit less pronounced than in CC, still underscores the need for their standardization and optimization (i.e., target muscles and optimal BoNT dose) to maximize the patients’ benefits and to provide recommendations for extensive use of BoNT to treat PS in clinical practice.

The slight improvement in AC and related pain among PD patients receiving BoNT treatment indicates a potential role for BoNT in managing anterior neck flexion [83,89]. However, the limited number of studies suggests that further research is necessary to define the most effective strategies for these specific APAs.

Only two studies addressed the effects of combined BoNT treatment with physical rehabilitative interventions, suggesting the potential synergistic effects of integrating multiple pharmacological and non-pharmacological treatment modalities [87,96]. The RCT study by Tassorelli et al. reported decreased APAs severity and pain alongside reduced disability and increased quality of life [87]. This supports the notion that a multidisciplinary approach may be more effective in managing APAs in PD. Difficulties in understanding the pathophysiology of these disorders have strongly influenced the development of specific rehabilitative approaches for APA in PD (as well as the use of BoNT in clinical practice). However, advancements in our understanding have fueled a growing interest in neurorehabilitation, as evidenced by the literature [97,98,99]. Initially, this interest manifested through case reports [96] or observational studies but it has since evolved towards the execution of methodologically robust RCTs [87]. This underscores the importance of comprehensive care strategies to address the complex needs of PD patients. Grounded in this theoretical framework, the rehabilitation of pathological anterior trunk flexion should concentrate on stabilization efforts to foster unconscious self-correction and ensure trunk stability during daily activities [97]. The most promising rehabilitation strategies enhance sensorimotor integration and bolster both feedforward mechanisms and cognitive aspects of postural control, including dual-tasking capabilities [97,98,99]. These objectives can also be met through aquatic therapies, leveraging the advantages of a microgravity environment to enhance sensorimotor integration processes and modulate muscle tone [99].

The variability in response to BoNT treatment across different types of PA in PD underscores the complexity of these conditions and the need for personalized treatment plans. Following the optimization of dopaminergic therapy and physiotherapy, the administration of BoNT injections into hyperactive axial muscles, as identified via EMG assessments, may be considered for carefully selected patients with PS [19]. It is critical to avoid administering injections into compensatory paraspinal and non-paraspinal muscles to prevent exacerbating the condition. The diversity of study designs and outcome measures across the literature complicates the ability to definitively conclude on the efficacy of BoNT for APAs in PD. However, the evidence suggests potential benefits, particularly in combination with rehabilitative interventions, that warrant further investigation. The multifaceted nature of pain in PD complicates the identification and implementation of effective therapies, as it necessitates a nuanced understanding and approach to treatment [13,14,67,80,100].

APAs pose a significant challenge in the management of PD. The progressive nature of APAs often results in abnormal stresses on the body’s joints, muscles, and ligaments, leading to significant nociceptive pain [17,18,101]. Note that neuropathic and nociplastic pain may also coexist in PD, further complicating the clinical picture [17,67,102].

Insights from our systematic review shed light on the intricate dynamics between APAs and pain within PD, highlighting the beneficial impact of BoNT combined with rehabilitation interventions to correct musculoskeletal paraspinal and non-paraspinal imbalances, improve posture alignment, and obtain pain relief [35,83,84,87,88,89,90,93,94,95,96]. This finding underscores the dual benefits of BoNT and physical rehabilitation in enhancing postural alignment and stability and reducing pain, reinforcing the necessity for multidimensional treatment [13,80]. Establishing this link emphasizes the critical need for integrated approaches in treatment planning [67,81].

Future research should focus on standardizing the EMG investigation to detect the hyperactive muscles and the injection techniques and outcome measures to facilitate comparison across studies. Additionally, more significant, RCTs are needed to better understand the long-term effects of BoNT treatment, alone and in combination with other interventions, on APAs in PD. Understanding the mechanisms behind the variable responses to BoNT will be crucial in optimizing treatment protocols to improve patient outcomes.

Overall, our review highlights the potential of BoNT as a component of a multidisciplinary approach to managing APAs in PD while also emphasizing the need for further research to refine treatment strategies and improve patient multidisciplinary care [2,19,103].

First, all healthcare professionals caring for PD patients (including neurologists, physiatrists, physical therapists, and occupational therapists) should use a unified set of assessment tools and agree upon standardized criteria for evaluation and treatment planning [1,27,104,105]. This collaborative approach ensures consistency and coherence in assessing and managing PA across different specialties.

Secondly, it is crucial to include patients in a comprehensive, multidisciplinary management plan. This plan should be grounded in evidence-based medical treatments (such as optimizing PD therapy and administering BoNT for eligible patients) alongside tailored rehabilitation programs. These programs must address each patient’s unique type and severity of APA, ensuring a personalized and effective intervention strategy from the outset [19].

Third, there is clear promise in the role of BoNT for managing APAs associated with PD. In particular, PS patients might benefit from BoNT injections under EMG (and possibly ultrasound) guidance. Also, CC can be improved in some cases. However, the different patterns of muscle hyperactivity and the differentiation between primary and compensatory hyperactivity remain a significant challenge for the correct targeting at a single-subject level.

The primary limitation of this systematic review is its broad period, which could impact intervention quality. Additionally, the significant outcome heterogeneity limits the feasibility of a meta-analytic assessment of treatment effects. This review is particularly relevant for advancing the field for several reasons. Firstly, it offers a comprehensive overview of the effects and safety of BoNT injections as a treatment modality for different types of APAs in PD, categorized by the updated classification [26]. This is vital because the efficacy of BoNT could fluctuate based on the severity of APAs, which, as elaborated, entails intricate interactions between central and peripheral element. Identifying specific dystonic patterns or muscle groups affected in different APAs could enable clinicians to tailor BoNT injection strategies more precisely, improving treatment efficacy and reducing side effects.

Moreover, this review highlights gaps in the current research and the need for longitudinal studies to assess the long-term effects of BoNT treatment on APAs in PD. Despite the promise of BoNT injections in managing symptoms, the evidence of BoNT efficacy needs to be more comprehensive, with a need for high-quality, long-term studies to better understand the optimal use of BoNT, including dosing, injection sites, and frequency of treatment. This would not only help establish standardized treatment protocols but also understand the progression of APAs over time and the potential for BoNT to alter their trajectory.

Understanding the role of BoNT within the broader context of patient management can guide comprehensive care strategies that address the multifaceted challenges faced by individuals with PD.

## 4. Conclusions and Future Directions

The BoNT’s comprehensive benefits, encompassing its effects on muscle hyperactivity and pain, underscore its potential as a multifaceted therapeutic tool in PD. However, the limitations of the current literature studies in terms of study design, methods, and variability of outcome measures, it is not possible to recommend a final recommendation for using BoNT in clinical practice.

BoNT injections targeting hyperactive muscles should be given careful consideration, with decisions based on EMG findings and made under the guidance of experienced professionals. It is crucial to refrain from injecting compensatory paraspinal and non-paraspinal muscles. Moreover, prompt physical rehabilitation is necessary to strengthen weak/compensatory trunk/neck muscles and improve balance and mobility. Managing APAs in PD patients is best achieved through an interdisciplinary approach implemented early on.

## 5. Materials and Methods

The systematic review protocol was recorded in the PROSPERO database with registration number CRD42023414769 and strictly followed the Preferred Reporting Items for Systematic Reviews and Meta-Analyses (PRISMA) guidelines during the entire review procedure [106]. Assessment of quality and evidence levels utilized the study quality assessment tools and the 5-item Oxford CEBM scale, respectively, accessed at https://www.nhlbi.nih.gov/health-topics/study-quality-assessment-tools on 5 February 2024 [107].

### 5.1. Selection Criteria for Studies

Our inclusion criteria encompassed all study designs involving participants with PD and APAs, with APAs defined following consensus-based nosology and specific cutoff criteria [26]. We specifically included studies that treated PD patients experiencing APAs with BoNT injections, either as a standalone therapy or in conjunction with rehabilitation. Conversely, we excluded studies involving the use of other interventions applied singly or combined (i.e., only oral pharmacology, injection therapies other than BoNT, deep brain stimulation, spinal surgery, standalone rehabilitative interventions, and orthoses), treatment outcomes different from postural abnormalities severity (i.e., gait and balance) or other reasons (i.e., different study population or non-pertinent studies). Excluded from our review were non-human trials, reviews, abstracts, conference proceedings, and protocol papers that lacked data collection.

### 5.2. Outcomes

The primary outcome assessed was the severity of the APA, utilizing clinical or instrumental assessment tools. We aimed to classify and report APAs according to the newly established Movement Disorder Society (MDS) Task Force Consensus criteria whenever possible [26]. Secondary outcomes encompassed evaluations of additional motor and non-motor symptoms, including those assessed using the MDS-Unified Parkinson’s Disease Rating Scale, Berg Balance Scale, and pain measurements.

### 5.3. Search Strategy

One author searched the MEDLINE (PubMed), Web of Science (all databases), and SCOPUS electronic databases from their inception until December 2023. The search terms “Parkinson’s disease”, “Multiple System Atrophy”, “Progressive Supranuclear Palsy”, “Dementia”, “Lewy body”, “Corticobasal syndrome”, “Axial manifestation”, and “Postural syndrome” paired with “Therapeutics”, “Therapy”, and “rehabilitation” were used in combination. This broad search strategy ensured the most comprehensive perspective on the topic, considering the various nosologies of postural abnormalities, their different manifestations, and interventions across time. The search strategy for all databases is detailed in the Appendix A.

### 5.4. Data Collection and Analysis

Three reviewers independently evaluated the relevance of studies based on titles and abstracts using the Rayyan online platform. Full-text articles were then obtained and independently assessed for inclusion. Any discrepancies were resolved through discussion. Subsequently, a separate group of three independent reviewers extracted data from the eligible full-text documents. They utilized an electronic data extraction form to document various details, divided into seven sections as follows: (1) author and publication year, type of study, and type of APAs; (2) number of patients, age at evaluation and PD duration; (3) APAs duration, diagnostic criteria of APAs, severity of APAs; (4) type of botulinum toxin, injection methods and number of injection cycles; (5) muscle injected; (6) study outcomes and follow-ups, and (7) key findings. Interventions were categorized as pharmacological injections with botulinum toxin, with or without accompanying rehabilitation.

## Figures and Tables

**Figure 1 toxins-16-00228-f001:**
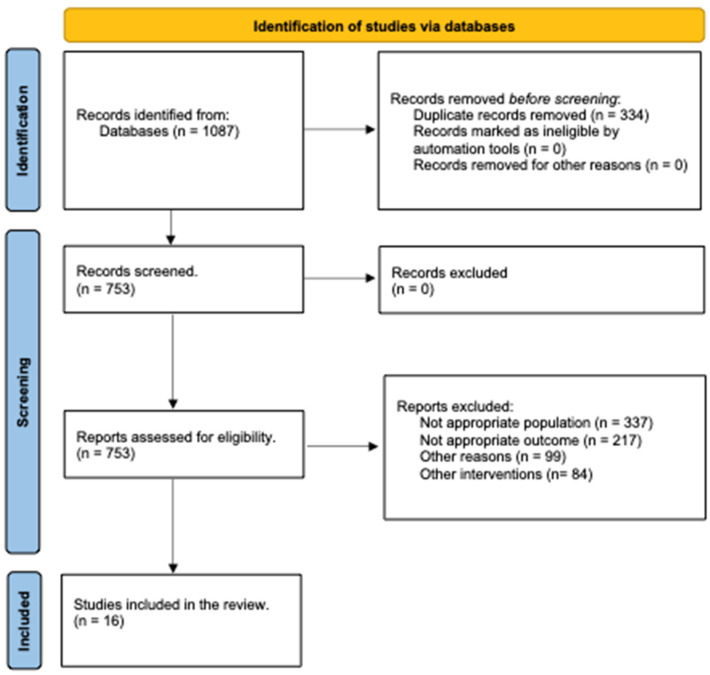
PRISMA flowchart for search strategy results.

**Figure 2 toxins-16-00228-f002:**
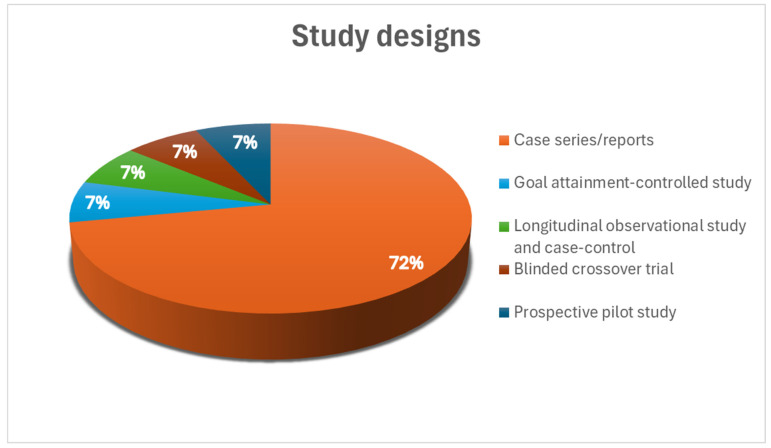
A pie chart describing the percentage and type of study design of BoNT treatment alone investigations in patients with APAs.

## Data Availability

This study did not create or analyze new data, and data sharing is not applicable to this article.

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
