# Peer review of "Botulinum Toxin for Axial Postural Abnormalities in Parkinson’s Disease: A Systematic Review"

_toxins, 2024, doi:10.3390/toxins16050228_

Round 1

Reviewer 1 Report

Comments and Suggestions for Authors

The study is written in good English. I have concerns that of 1087 records identified, only 16 met their eligibility criteria, some being rejected as "not appropriate outcome." Nevertheless 6 of these remaining studies included in this review were labelled as poor quality. The opening paragraph of the results section is slightly confusing, as it is not entirely clear whether the description of the studies is only about the included studies.

Table 1 is very difficult to read and understand. Such individual details would be better presented in an Appendix with the patients grouped together, giving clear criteria for such grouping. In addition it would be better presented in landscape form, perhaps in a smaller font, and the headings should be repeated on each fresh page.

Sections 2.1 to 2.5 are entirely descriptive with minimal attempt to group together similar treatments and outcomes, and thus it is hard to follow whether there were any therapeutic benefits. In the discussion, even whether the authors admit there were varying outcomes, they appear to focus on those that were positive. Given the quality of the data they were considering, it is difficult to believe that they are able to draw any firm conclusion, apart from, as they say, the need for "high-quality, long-term studies."

Author Response

Reviewer 1

Comments to the Author.

The study is written in good English. I have concerns that of 1087 records identified, only 16 met their eligibility criteria, some being rejected as "not appropriate outcome." Nevertheless 6 of these remaining studies included in this review were labelled as poor quality. The opening paragraph of the results section is slightly confusing, as it is not entirely clear whether the description of the studies is only about the included studies.

Authors reply.

Thank you for your feedback on the selection criteria and the opening paragraph of the results section. In the methods section, we clarified the reasons for excluding records reported in Figure 1. In detail, we excluded studies involving the use of other interventions applied singly or combined (i.e., only oral pharmacology, injection therapies other than BoNT, deep brain stimulation, spinal surgery, standalone rehabilitative interventions, and orthoses), treatment outcomes different from postural abnormalities severity (i.e., gait and balance) or other reasons (i.e., different study population or not pertinent studies). Additionally, we used a broad search strategy that took into account the heterogeneity in defining diseases and treatments over the years, thus aiming to have a broad and comprehensive perspective. This explains why we found a large number of records despite a limited number of papers included.

We have improved the readability and clarity of the opening paragraph of the results section. Additionally, we have clarified that the 16 records mentioned were selected according to the review's selection criteria. Table 1 summarizes the leading information from these included studies. To keep Table 1 concise and straightforward, we have included detailed information about the studies in the Table 1 appendix.

Comments to the Author.

Table 1 is very difficult to read and understand. Such individual details would be better presented in an Appendix with the patients grouped together, giving clear criteria for such grouping. In addition, it would be better presented in landscape form, perhaps in a smaller font, and the headings should be repeated on each fresh page.

Authors reply.

Thank you for your feedback. The leading information from the included studies is now summarized in Table 1, presented in landscape form. It emphasizes positive or negative findings for each type of postural abnormality and provides the main reference to the Appendix. All the remaining detailed information has been placed in the Appendix. This organization ensures that Table 1 remains clear and easy to understand, but the reader can find all the remaining study details in the Table 1 Appendix.

Comments to the Author.

Sections 2.1 to 2.5 are entirely descriptive with minimal attempt to group together similar treatments and outcomes, and thus it is hard to follow whether there were any therapeutic benefits.

Authors reply.

Thank you for your feedback. Sections 2.1 and 2.5 have been revised per your suggestion, maintaining the main results in the text and moving all the remaining detailed information in the Table 1 Appendix. A new figure (Figure 2) has been added to visually represent the experimental designs' heterogeneity and to lighten the text of the cited sections.

Comments to the Author.

In the discussion, even whether the authors admit there were varying outcomes, they appear to focus on those that were positive. Given the quality of the data they were considering, it is difficult to believe that they are able to draw any firm conclusion, apart from, as they say, the need for "high-quality, long-term studies."

Authors reply.

Thank you for your comment. As acknowledged at the beginning of the discussion, we referred to positive/negative main findings mentioning the variability in outcomes and the limitations inherent in the data quality from the studies reviewed. We conclude that the limitations of the current literature studies, in terms of study design, methods, and variability of outcome measures, do not allow drawing final recommendation for using BoNT in clinical practice.

Reviewer 2 Report

Comments and Suggestions for Authors

In this paper authors have review papers on influence of BoNT treatment for postural abnormalities in patients with parkinsonism. Introduction part is very well described. Research flow with keywords and data base involved in research are also described. Results and discussion are clearly written. Conclusions are made based on research results. References are adequate. In manuscript there are nothing else that should be changed. I only have comment on Table 1 transparency. It is hard to find data in this table. So, I ask authors to change table to be more clearly readable.

Author Response

Reviewer 2

In this paper authors have review papers on influence of BoNT treatment for postural abnormalities in patients with parkinsonism. Introduction part is very well described. Research flow with keywords and data base involved in research are also described. Results and discussion are clearly written. Conclusions are made based on research results. References are adequate. In manuscript there are nothing else that should be changed. I only have comment on Table 1 transparency. It is hard to find data in this table. So, I ask authors to change table to be more clearly readable.

Authors reply.

Thank you for your concerns on Table 1. Accordingly, the leading information from the included studies has been summarized in Table 1 presented in landscape form. It now emphasizes positive or negative findings for each type of postural abnormality and provides the main reference to the Appendix. All the remaining detailed information has been placed in the Appendix. This organization ensures that Table 1 remains clear and easy to understand, but the reader can find all the remaining study details in the Table 1 Appendix.

Reviewer 3 Report

Comments and Suggestions for Authors

In this review, authors described the current research findings on therapeutic potential of Botulinum Toxins for the treatment of axial postural abnormalities (APAs) in patients suffering from Parkinson’s disease (PD).

The manuscript is well-written, and applied method for this report is acceptable. Authors highlighted on the beneficial impact of BoNT injections combined with rehabilitation interventions in treating APAs in PD patients and recommended for integrated approaches in treatment planning for the individual patients.

This reviewer noticed the following minor points.

1.   Layout of Table 1 should be in landscape because current setting is difficult to follow by the readers.

2.  Line 158, “Table 2” should be checked and corrected.  Only Table 1 was described for 16 records included in this review.

Author Response

Reviewer 3

In this review, authors described the current research findings on therapeutic potential of Botulinum Toxins for the treatment of axial postural abnormalities (APAs) in patients suffering from Parkinson’s disease (PD). 

The manuscript is well-written, and applied method for this report is acceptable. Authors highlighted on the beneficial impact of BoNT injections combined with rehabilitation interventions in treating APAs in PD patients and recommended for integrated approaches in treatment planning for the individual patients. 

This reviewer noticed the following minor points.

  1. Layout of Table 1 should be in landscape because current setting is difficult to follow by the readers.
  2. Line 158, “Table 2” should be checked and corrected.  Only Table 1 was described for 16 records included in this review.

Authors reply.

  1. Thank you for your concerns on Table 1. Accordingly, the leading information from the included studies has been summarized in Table 1 and presented in landscape form. It now emphasizes positive or negative findings for each type of postural abnormality and provides the main reference to the Appendix. All the remaining detailed information has been placed in the Table 1 Appendix. This organization ensures that Table 1 remains clear and easy to understand, but the reader can find all the remaining study details in the Table 1 Appendix.
  2. The typos have been corrected.